# An Hybrid Integration Method-Based Track-before-Detect for High-Speed and High-Maneuvering Targets in Ubiquitous Radar

**Xiangyu Peng** , **Qiang Song** , **Yue Zhang *** and **Wei Wang**

School of Electronic and Communication Engineering, Sun Yat-sen University, Shenzhen 518107, China; pengxy57@mail2.sysu.edu.cn (X.P.); songq9@mail2.sysu.edu.cn (Q.S.); wangw278@mail.sysu.edu.cn (W.W.)
* Correspondence: zhangyue8@mail.sysu.edu.cn

**Abstract:** Due to the limited transmission gain of ubiquitous radar systems, it has become necessary to use a long-time coherent integration method for range-Doppler (RD) analysis. However, when the target exhibits high-speed and high-maneuver capabilities, it introduces challenges, such as range migration (RM), Doppler frequency migration (DFM), and velocity ambiguity (VA) in the RD domain, thus posing significant difficulties in target detection and tracking. Moreover, the presence of VA further complicates the problem. To address these complexities while maintaining integration efficiency, this study proposes a hybrid integration approach. First, methods called Keystone-transform (KT) and matched filtering processing (MFP) are proposed for compensating for range migration (RM) and velocity ambiguity (VA) in Radar Detection (RD) images. The KT approach is employed to compensate for RM, followed by the generation of matched filters with varying ambiguity numbers. Subsequently, MFP enables the production of multiple RD images covering different but contiguous Doppler frequency ranges. These RD images can be compiled into an extended RD (ERD) image that exhibits an expanded Doppler frequency range. Second, an improved particle-filter (IPF) algorithm is raised to perform incoherent integration among ERD images and to achieve track-before-detect (TBD) for a target. In the IPF, the target state vector is augmented with ambiguous numbers, which are estimated via maximum posterior probability estimation. Then, to compensate for the DFM, a line spread model (LSM) is proposed instead of the point spread model (PSM) used in traditional PF. To evaluate the efficacy of the proposed method, a radar simulator is devised, encompassing comprehensive radar signal processing. The findings demonstrate that the proposed approach achieves a harmonious equilibrium between integration efficiency and computational complexity when it comes to detecting and tracking high-speed and high-maneuvering targets with intricate maneuvers. Furthermore, the algorithm's effectiveness is authenticated by exploiting ubiquitous radar data.

**Keywords:** ubiquitous radar; hybrid integration method; keystone transform; matched filtering processing; improved particle filter; track-before-detect

## 1. Introduction

Ubiquitous radar is also referred to as holographic staring radar or floodlight radar [1–3]. In 2017, the IEEE Radar Definition Standard gave the first definition of ubiquitous radar, which is a radar that looks everywhere all the time and performs multiple functions simultaneously instead of sequentially [4]. Ubiquitous radar employs a wide-beam transmitting antenna to uniformly irradiate a wide area with transmitted signal energy. On the receiving end, a digital array antenna incorporates digital beam forming (DBF) technology to obtain multiple narrow beams that cover the entire transmitting area simultaneously. This enables uninterrupted monitoring and detection of the entirety of the airspace under surveillance. Ubiquitous radar adopts the long-time coherent accumulation (LTCA) technique [5], which can improve the radar transmission power utilization, enhance the detection performance, and improve the target information acquisition ability under the same power aperture

conditions. However, during the LTCA process, the motion characteristics of the high-speed and high-maneuvering target result in range migration (RM), Doppler frequency migration (DFM), and velocity ambiguity (VA), which can degrade the performance of coherent accumulation and seriously affect the subsequent tracking.

Currently, extensive research is being conducted on compensating for target movement in the LTCA process. In this regard, Keystone transform (KT) has emerged as a widely used method for compensating reference motion (RM) [6–8]. Additionally, both Radon Fourier Transform (RFT) [9–11] and Generalized Radon Fourier Transform (GRFT) [12] have been proposed to compensate for first-order RM and high-order RM, respectively. Furthermore, numerous algorithms, such as the Keystone-Transform and Lv's distribution (KT-LVD) [13], Radon-Lv's distribution (RLVD) [14], and Radon-Fractional Fourier Transform (RFRFT) [15], have been developed to compensate for RM and Doppler frequency modulation (DFM) in targets exhibiting uniformly accelerated motion. These methodologies offer invaluable remedies for mitigating the detrimental impact of motion-induced artifacts in the LTCA process. Nevertheless, it is essential to note that these computational algorithms necessitate a substantial amount of processing power. There are some hybrid integration algorithms, consisting of coherent and incoherent integration, that can reduce computational complexity, but they often come at the cost of significantly reducing integration gain [16–18].

The track-before-detect (TBD) method is capable of achieving incoherent integration. This method involves exploiting a signal processing algorithm to perform detections even when there is significant noise interference. Additionally, TBD establishes tracks for potential targets even before they are officially detected. This technique is used to improve the detection performance of radar systems in situations where the target signal is weak or difficult to distinguish from surrounding clutter or interference [19,20]. In contrast to the detect-before-track algorithm, which is a popular technique to track targets after the detection process [21], TBD does not need threshold the raw data for detection, it can operate detection and tracking jointly on the raw data [22]. Furthermore, the TBD can achieve a more robust detection via tracking all data up to the current time step, instead of depending solely on the current time step [23].

Numerous algorithms are available for implementing the TBD, which can be broadly classified into two categories: batch algorithms and recursive algorithms. Batch algorithms, such as the Hough transform (HT) algorithm [24,25] and dynamic programming (DP) algorithm [26–28], store and then process received data within a certain period of time, which implies that several scan times are required for making a more accurate result. This feature determines that these algorithms cannot be adapted to a maneuvering target. Furthermore, as a type of batch algorithm, the multi-frame (MF) algorithm can be applied for the TBD of maneuvering mobile targets [29–31].

Recursive algorithms predict and then update the estimation of the target state recursively, which is perfectly consistent with the full space and time domain coverage property of ubiquitous radar. As a typical recursive algorithm, a particle filter (PF) is usually chosen for implementing the TBD approach [32–35], as it provides a significant advantage for non-linear and non-Gaussian filtering problems. PF-TBD was first proposed by Salmond [36], enjoying several advantages over previous approaches. In Salmond's PF method, a variable is added to represent the target existence state into the state vector, which can cause the degradation of the particle group. Rutten [37] has improved the PF by formulating the target existence probability more efficiently.

The objective of this paper is to explore the application of a hybrid integration technique for effectively achieving simultaneous detection and tracking of a weak and maneuvering target in ubiquitous radar. The compensation for RM and VA occurs during coherent integration, while compensating for DFM occurs during incoherent integration. During the coherent integration step, compensation for the first-order RM is achieved by employing KT, followed by the application of matched filtering processing (MFP) to mitigate the effects arising from VA. In this step, each ambiguity number can be utilized to generate

a corresponding matched filter. The use of different matched filters enables the creation of sub-RD images with distinct yet contiguous Doppler frequency ranges. These sub-RD images can then be combined to form an extended RD (ERD) image encompassing the entire Doppler frequency spectrum. In the incoherent integration step, an improved particle filter (IPF) is proposed and applied in ERD images to realize the TBD. In this approach, the IPF incorporates an ambiguous number in the target state vector, which can be estimated using maximum posterior probability. Furthermore, to compensate for the DFM, a line spread model (LSM) is proposed as a replacement for the traditional point spread model (PSM) used in conventional PFs [37]. The proposed method is named KT-MFP-IPF-TBD in this paper.

The experiments show that the IPF-TBD method has better detection and tracking performance compared to the traditional PF-TBD method, and The integration gain of the KT-MFP-IPF-TBD method is close to LTCA, but its computational complexity is much lower than the LTCA.

The rest of this paper is organized as follows. Section 2 shows the deduction of ERD-image generation, including RM correction and VA compensation. Section 3 describes the IPF algorithm based on the Bayesian theory and presents a major implementation of the filter. Next, the simulation and measurement results are displayed and discussed in Section 4. Section 5 presents our conclusion.

## 2. Keystone-Transform and Matched Filter Processing

The transmission signal employed by the commonly used ubiquitous radar is characterized as a linear frequency modulated (LFM) signal.

$$
\begin{aligned}
s(t, t_m) = rect\left(\frac{t - t_m}{T_p}\right) \\
\times \exp\left[j\left(2\pi f_c(t - t_m) + \pi\gamma(t - t_m)^2\right)\right],
\end{aligned}
\tag{1}
$$

where $T_p$ is the pulse width, $\gamma$ is the frequency modulated rate, $f_c$ is the carrier frequency, $t_m = mT_r\ m = 0, 1, ..., M - 1$ is the slow time, $T_r$ denotes the pulse repetition interval (PRI), $M$ is the number of coherent integrated pulses, $t$ is the fast time, and

$$
rect(u) = \left\{
\begin{array}{ll}
1 & |u| \leq \frac{1}{2} \\
0 & |u| > \frac{1}{2}
\end{array}
\right.
\tag{2}
$$

is a rectangle window function.

The received baseband signal can be stated as

$$
\begin{aligned}
s_1(\hat{t}, t_m) = A_0 rect\left(\frac{\hat{t} - \frac{2R(t_m)}{c}}{T_p}\right) rect\left(\frac{t_m}{T_I} - \frac{1}{2}\right) \\
\times \exp\left[j\left(4\pi f_c \frac{R(t_m)}{c} + \pi\gamma\left(\hat{t} - \frac{2R(t_m)}{c}\right)^2\right)\right],
\end{aligned}
\tag{3}
$$

where $c$ is the light speed, $A_0$ is the amplitude, $\hat{t} = t - t_m$ represents new fast time and $R(t_m)$ is the instantaneous range between radar and target. $T_I$ is the integration time and is defined as

$$
T_I = MT_r.
\tag{4}
$$

Assume that there is a moving target given an initial radial range $R_0$ at $t_m = 0$. Therefore the instantaneous radial range $R(t_m)$ between the radar and a maneuvering target with constant acceleration can be written as

$$
R(t_m) = R_0 + v_o t_m + \frac{1}{2}a_0 t_m^2,
\tag{5}
$$

where $v_o$ and $a_0$ are defined as the radial velocity and acceleration of the target, respectively.

After the pulse compression in the range-frequency domain, the compressed signal can be expressed as

$$S_1(f, t_m) = A_1 rect\left(\frac{f}{B}\right) rect\left(\frac{t_m}{T_I} - \frac{1}{2}\right) \times \exp\left[-j4\pi(f + f_c)\frac{R(t_m)}{c}\right], \tag{6}$$

where $A_1$ is the amplitude and $B$ is the bandwidth of the transmitted signal. Then, inserting (5) into (6), we have

$$\begin{aligned} S_1(f, t_m) &= A_1 rect\left(\frac{f}{B}\right) rect\left(\frac{t_m}{T_I} - \frac{1}{2}\right) \times \exp\left[-j4\pi(f + f_c)\frac{R_0}{c}\right] \\ &\quad \times \exp\left[-j4\pi(f + f_c)\frac{v_o t_m}{c}\right] \times \exp\left[-j2\pi(f + f_c)\frac{a_0 t_m^2}{c}\right]. \end{aligned} \tag{7}$$

However, for a fast-moving target, its Doppler frequency may exceed the pulse repetition frequency (PRF), according to [5], $v_o$ can be written as

$$v_o = v_{base} + \frac{\lambda \cdot PRF}{2}\eta, \tag{8}$$

where $\lambda$ is the wavelength and is defined as $\lambda = c/f_c$. $v_{base}$ represents the baseband velocity and it satisfies $v_{base} \in [-\lambda \cdot PRF/4, \lambda \cdot PRF/4]$. $\eta$ denotes the ambiguity number. After substituting (8) into (7), $S_1(f, t_m)$ can be rewritten as

$$\begin{aligned} S_1(f, t_m) &= A_1 rect\left(\frac{f}{B}\right) rect\left(\frac{t_m}{T_I} - \frac{1}{2}\right) \times \exp\left[-j4\pi(f + f_c)\frac{R_0}{c}\right] \\ &\quad \times \exp\left[-j4\pi(f + f_c)\frac{v_{base} t_m}{c}\right] \times \exp\left[-j2\pi(f + f_c)\frac{a_0 t_m^2}{c}\right] \\ &\quad \times \exp\left[-j2\pi(f + f_c)\eta\frac{\lambda \cdot PRF}{c}t_m\right]. \end{aligned} \tag{9}$$

Consider the following equation:

$$\exp\left(-j2\pi f_c \eta\frac{\lambda \cdot PRF}{c}t_m\right) = \exp(-j2\pi\eta m) = 1. \tag{10}$$

$S_1(f, t_m)$ can be further simplified as

$$\begin{aligned} S_1(f, t_m) &= A_1 rect\left(\frac{f}{B}\right) rect\left(\frac{t_m}{T_I} - \frac{1}{2}\right) \times \exp\left[-j4\pi(f + f_c)\frac{R_0}{c}\right] \\ &\quad \times \exp\left[-j4\pi(f + f_c)\frac{v_{base} t_m}{c}\right] \times \exp\left[-j2\pi(f + f_c)\frac{a_0 t_m^2}{c}\right] \\ &\quad \times \exp\left(-j2\pi f\eta\frac{\lambda \cdot PRF}{c}t_m\right). \end{aligned} \tag{11}$$

In a number of works [38–40], the expression of KT is denoted as

$$t_m = \frac{f_c}{f_c + f}\tau_m, \tag{12}$$

where $\tau_m$ denotes the new slow-time variable.

To eliminate the coupled term between $f$ and $t_m$, we apply KT to (11),

$$
\begin{aligned}
S_1(f, \tau_m) \approx{} & A_1 rect\left(\frac{f}{B}\right) rect\left(\frac{\tau_m}{T_I} - \frac{1}{2}\right) \times \exp\left[-j4\pi(f + f_c)\frac{R_0}{c}\right] \\
& \times \exp\left[-j4\pi\frac{v_{base}\tau_m}{\lambda}\right] \times \exp\left[-j2\pi\frac{f_c^2}{(f + f_c)}\frac{a_0\tau_m^2}{c}\right] \\
& \times \exp\left(-j2\pi\eta\frac{f}{f_c + f}\tau_m \cdot PRF\right).
\end{aligned}
\tag{13}
$$

Because $f_c \gg f$, we can obtain

$$
\begin{aligned}
\frac{f_c}{f_c + f} &\approx 1 \\
\frac{\hat{f}}{f_c + f} &\approx \frac{f}{f_c}
\end{aligned}.
\tag{14}
$$

Therefore, (13) can be simplified as

$$
\begin{aligned}
S_1(f, \tau_m) ={} & A_1 rect\left(\frac{f}{B}\right) rect\left(\frac{\tau_m}{T_I} - \frac{1}{2}\right) \times \exp\left[-j4\pi(f + f_c)\frac{R_0}{c}\right] \\
& \times \exp\left(-j4\pi\frac{v_{base}\tau_m}{\lambda}\right) \times \exp\left(-j2\pi\frac{a_0\tau_m^2}{\lambda}\right) \times \exp\left(-j2\pi\eta\frac{f}{f_c}m\right).
\end{aligned}
\tag{15}
$$

According to (15), it can be inferred that despite the successful compensation of first-order RM, there still remains a linear RM induced by the ambiguity velocity (as indicated by the final term). Therefore, we can construct a matched filter to solve the residual coupled terms between $f$ and $\eta$.

$$
H_{match}^{\hat{\eta}}(f, m) = \exp\left(j2\pi\frac{f}{f_c}\hat{\eta}m\right)
\tag{16}
$$

where $\hat{\eta}$ is the estimated value of the ambiguity number. After multiplying $H_{match}^{\hat{\eta}}(f, m)$ by $S_1(f, t_m)$ in (15), one has

$$
\begin{aligned}
S_2^{\hat{\eta}}(f, \tau_m) ={} & S_1(f, \tau_m) \cdot H_{match}^{\hat{\eta}}(f, m) \\
={} & A_1 rect\left(\frac{f}{B}\right) rect\left(\frac{\tau_m}{T_I} - \frac{1}{2}\right) \times \exp\left[-j4\pi(f + f_c)\frac{R_0}{c}\right] \\
& \times \exp\left(-j2\pi(\eta - \hat{\eta})\frac{f}{f_c}m\right) \times \exp\left(-j4\pi\frac{v_{base}\tau_m}{\lambda}\right) \times \exp\left(-j2\pi\frac{a_0\tau_m^2}{\lambda}\right).
\end{aligned}
\tag{17}
$$

When $\eta = \hat{\eta}$, the VA can be compensated.

$$
\begin{aligned}
S_2^{\eta}(f, \tau_m) ={} & A_1 rect\left(\frac{f}{B}\right) rect\left(\frac{\tau_m}{T_I} - \frac{1}{2}\right) \times \exp\left[-j4\pi(f + f_c)\frac{R_0}{c}\right] \\
& \times \exp\left(-j4\pi\frac{v_{base}\tau_m}{\lambda}\right) \times \exp\left(-j2\pi\frac{a_0\tau_m^2}{\lambda}\right).
\end{aligned}
\tag{18}
$$

After performing the Inverse Fourier transform (IFT) of $f$ in $S_2^{\eta}(f, \tau_m)$, we can obtain

$$
\begin{aligned}
s_2(\hat{t}, \tau_m) ={} & A_2 rect\left(\frac{\tau_m}{T_I} - \frac{1}{2}\right) sinc\left(B\left(\hat{t} - \frac{2R_0}{c}\right)\right) \\
& \times \exp\left(-j4\pi\frac{\left(R_0 + v_{base}\tau_m + \frac{1}{2}a_0\tau_m^2\right)}{\lambda}\right),
\end{aligned}
\tag{19}
$$

where $A_2$ is the new amplitude. From (19), it shows that the combined product of $a_0$ and $\tau_m^2$ will introduce the DFM, which can lead to the spread of the target's Doppler. Next, we can calculate the Fourier transform (FT) of $\tau_m$ based on the stationary phase principle [41,42].

$$
\begin{aligned}
S_3\left(\hat{t}, \xi\right) &= FT_{\tau_m}\left(s_2\left(\hat{t}, \tau_m\right)\right) \\
&\approx A_3 rect\left(\frac{\lambda(\xi - \xi_0)}{2a_0 T_I}\right) \sin c\left(B\left(\hat{t} - \frac{2R_0}{c}\right)\right) \times \exp\left[j\pi \frac{\lambda}{2a_0}(\xi - \xi_0)^2\right] \\
&\times \exp\left(j\pi \frac{-4R_0}{\lambda} + j\frac{\pi}{4}\right),
\end{aligned}
\tag{20}
$$

where $A_2$ is the new amplitude. $\xi_0$ can be expressed as

$$
\xi_0 = -\frac{2v_{base}}{\lambda} - \frac{a_0 T_I}{\lambda}.
\tag{21}
$$

From (20), the width of DFM is given as

$$
Width_{(DFM)} = \frac{2a_0 T_I}{\lambda}.
\tag{22}
$$

In Figure 1, the RD images with DFM and without DFM are displayed. As the Doppler frequency of the target has broadened, the PSM is not applicable when there is DFM in RD image.

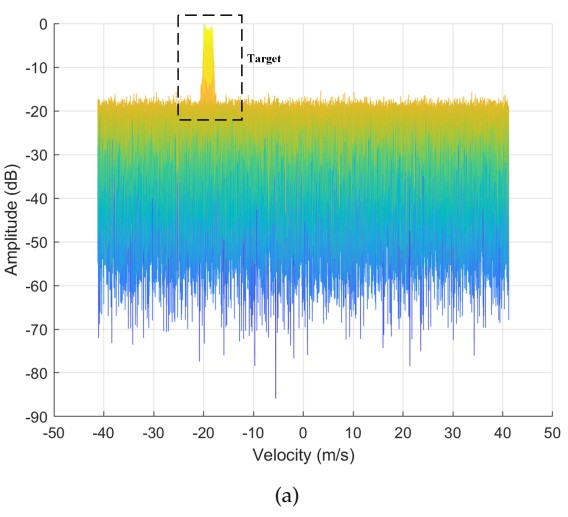

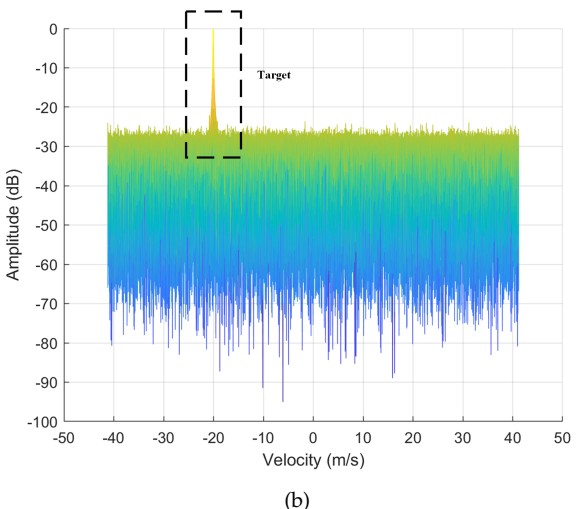

(a)                                                    (b)

**Figure 1.** Sub-RD imaging. (**a**) With DFM. (**b**) Without DFM.

According to the above description, each ambiguity number can produce a corresponding matched filter, different matched filters generate sub-RD images with different Doppler frequency ranges, which are defined as $\left[\frac{(2\eta - 1)\lambda \cdot PRF}{4}, \frac{(2\eta + 1)\lambda \cdot PRF}{4}\right]$. Since they are consecutive, we can stitch them together to obtain an ERD image, as Figure 2 shows. $\Psi$ represents the number of ambiguous numbers, which are odd. The velocity range of ERD images is given as $\left[\frac{-\Psi\lambda \cdot PRF}{4}, \frac{\Psi\lambda \cdot PRF}{4}\right]$. A simulation result of an ERD image with $\Psi = 3$ is shown in Figure 3. The substructures present in areas A, B, and C correspond to different values of $\eta$, specifically $\eta = -1$, $\eta = 0$, and $\eta = 1$, respectively. When the ambiguity number is matched (area B), RM is small. In contrast, when the ambiguity number is not matched (area A and area C), RM becomes very large.

Denote $T_s$ as the sampling interval, the length of pulse is $N = T_r/T_s$. Hence, the size of the ERD image is $N \times \Psi \cdot M$.

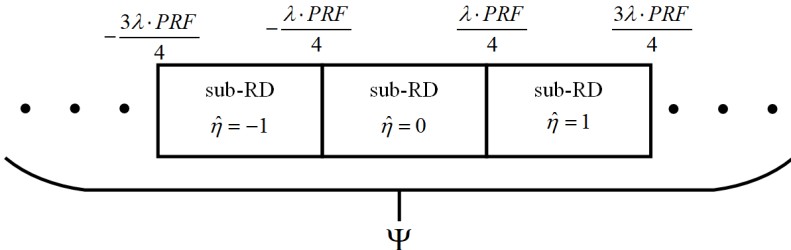

**Figure 2.** Demonstration of stitching RD images.

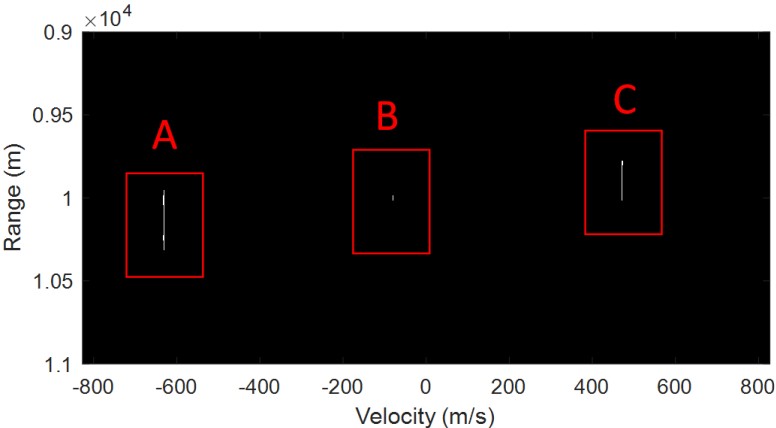

**Figure 3.** Simulation of an ERD image.

## 3. Improved Particle Filter

### 3.1. The Target Motion Model

In a tracking system, the performance of the filter can be guaranteed when the target model matches the actual target motion. In this paper, the state vector of a target is given as:

$$\widetilde{\mathbf{x}}_p = \left[\begin{array}{ccccc} r_p, & \eta_p, & v_p^{base}, & a_p, & I_p \end{array}\right]^T, \tag{23}$$

where $p$ is the time index, $\eta_p$ is ambiguity number, $r_p$ is radial distance, $v_p^{base}$ is the baseband radial velocity, $a_p$ is the radial acceleration and $I_p$ represents the intensity of the target at index $p$.

To simplify calculations, the state vector is rewritten as

$$\mathbf{x}_p = \left[\begin{array}{cccc} r_p, & v_p, & a_p, & I_p \end{array}\right]^T, \tag{24}$$

where $v_p = v_p^{base} + \eta_p v_{\max}$ and $v_{\max} = \frac{\lambda \cdot PRF}{2}$.

Then the state-transition function is

$$\mathbf{x}_p = \mathbf{F} \cdot \mathbf{x}_{p-1} + \mathbf{g} \cdot \mathbf{w}, \tag{25}$$

where $\mathbf{F}$ is the transition matrix and $\mathbf{g}$ is the corresponding process noise input matrix. They can be defined as

$$\mathbf{F} = \begin{bmatrix} 1 & T_I & \frac{1}{2}T_I^2 & 0 \\ 0 & 1 & T_I & 0 \\ 0 & 0 & 1 & 0 \\ 0 & 0 & 0 & 1 \end{bmatrix} \tag{26}$$

and

$$\mathbf{g} = \begin{bmatrix} \frac{1}{6}T_I^3 & \frac{1}{2}T_I^2 & T_I & 0 \\ 0 & 0 & 0 & 1 \end{bmatrix}^T. \tag{27}$$

**w** represents the statistically independent, two-dimensional white Gaussian noise with zeros mean and its covariance matrix is

$$\mathbf{Q} = \begin{bmatrix} q_1 & 0 \\ 0 & q_2 \end{bmatrix}. \tag{28}$$

The variable $q_1$ describes the model uncertainties in the state-transition equation. Target tracking typically uses the maximum acceleration rate as its value. $q_2$ is the power spectral density of the noise in the rate of change in the target reflection amplitude.

The presence of the target at time $p$ is described by a target existence variable $E_p$, $E_p = 1$ represents the target is alive, otherwise, $E_p = 0$. The probability of the target existence is modeled as a first-order Markov process and the corresponding probability transition matrix is

$$\Pi = \begin{bmatrix} 1 - P_d & P_b \\ P_d & 1 - P_b \end{bmatrix}, \tag{29}$$

where $P_b = P\{E_k = 1 | E_{k-1} = 0\}$ is the probability of target birth and $P_d = P\{E_k = 0 | E_{k-1} = 1\}$ represents the probability of target death. Furthermore, $P\{E_k = 1 | E_{k-1} = 1\}$ is continuing probability.

### 3.2. Observation Model

Following the above description, we use the ERD images as the observation data. The measurement sets are

$$\mathbf{z}_p = \left\{ z_p^{(n,k)}, k = 1, 2, ..., \Psi M; n = 1, 2, ..., N \right\}. \tag{30}$$

where $n$ and $k$ represent the index of the axes. $z_p^{n,k}$ is the value of the ERD image cell and can be expanded as

$$z_p^{(n,k)} = \begin{cases} I_p h^{(n,k)}(\mathbf{x}_p) + v_p^{(n,k)} \\ v_p^{(n,k)} \end{cases}. \tag{31}$$

where $h^{(n,k)}(\mathbf{x}_p)$ is a contribution from the intensity of the target in the cell $(n, k)$ and $v_p^{(n,k)}$ is the measurement noise in the cell, which is a known Gaussian distribution with zeros mean, variance $\sigma_v^2$. Some paper use the PSM as $h^{(n,k)}(\mathbf{x}_p)$ [19]. In this paper, we propose a new spread function named LSM expressed in (32) and (33) to compensate the DFM.

While $a_p \geq 0$

$$\begin{aligned} h^{(n,k)}(\mathbf{x}_p) = & \left| \sin c\left( \frac{2B}{c}(n\Delta_r - r_p) \right) \right| \times \left[ rect\left( \frac{k\Delta_v - v_p}{a_p T_I} - \frac{1}{2} \right) \right. \\ & + \exp\left[ -\frac{(k\Delta_v - v_p - a_p T_I)^2}{2\Sigma^2} \right] \cdot [k\Delta_v > v_p + a_p T_I] \\ & + \left. \exp\left[ -\frac{(k\Delta_v - v_p)^2}{2\Sigma^2} \right] \cdot [k\Delta_v < v_p] \right]. \end{aligned} \tag{32}$$

While $a_p < 0$

$$\begin{aligned} h^{(n,k)}(\mathbf{x}_p) = & \left| \sin c\left( \frac{2B}{c}(n\Delta_r - r_p) \right) \right| \times \left[ rect\left( \frac{k\Delta_v - v_p}{a_p T_I} - \frac{1}{2} \right) \right. \\ & + \exp\left[ -\frac{(k\Delta_v - v_p - a_p T_I)^2}{2\Sigma^2} \right] \cdot [k\Delta_v < v_p + a_p T_I] \\ & + \left. \exp\left[ -\frac{(k\Delta_v - v_p)^2}{2\Sigma^2} \right] \cdot [k\Delta_v > v_p] \right]. \end{aligned} \tag{33}$$

where $\Sigma$ is a known parameter that controls the blurring. $\Delta_r$ and $\Delta_v$ are the length of each ERD cell in the range and the velocity axes. Figure 4 shows a example of $h^{(n,k)}(\mathbf{x}_p)$.

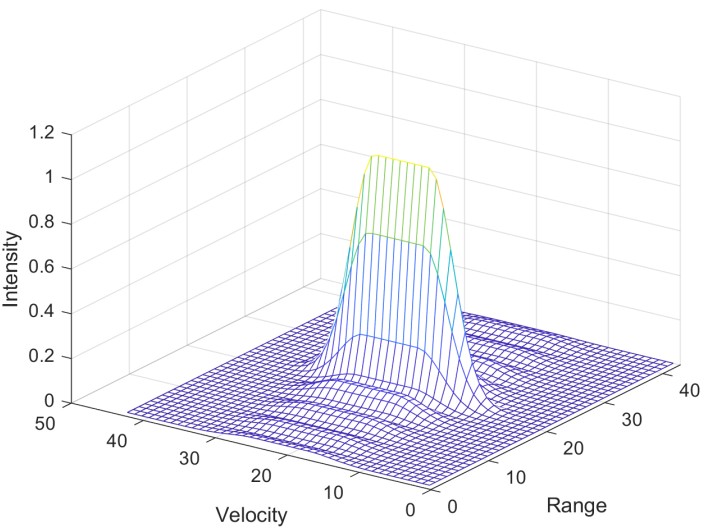

**Figure 4.** The line spread model.

### 3.3. Filter Derivation

The filter estimates $\mathbf{x}_p$ based on a set of all available measurements $\mathbf{z}_{1:p} = \{\mathbf{z}_i, i = 1, 2, ..., p\}$. In the Bayesian setting, the tracking problem can be transformed to a recursive approximation of the posterior probability density function (PDF) [43], $P(\mathbf{x}_p, E_k | \mathbf{z}_{1:p})$, and thus can be split into two stages: prediction and update. We use the PF method proposed by Rutten [37], where the target existence is separated from the target state.

$$
\begin{aligned}
P(\mathbf{x}_p, E_p = 1 | \mathbf{z}_{1:p}) &= P(\mathbf{x}_p | \mathbf{z}_{1:p}, E_p = 1) \\
&\times P(E_p = 1 | \mathbf{z}_{1:p}).
\end{aligned}
\tag{34}
$$

In (34), $P(\mathbf{x}_p | \mathbf{z}_{1:p}, E_p = 1)$ denotes the joint state densities and $P(E_p = 1 | \mathbf{z}_{1:p})$ is the probability of target existence.

The joint state densities can be expanded as

$$
\begin{aligned}
P(\mathbf{x}_p | \mathbf{z}_{1:p}, E_p = 1) &= \sum_{E_{p-1} = \{1, 0\}} P(E_{p-1} | \mathbf{z}_{1:p}, E_p = 1) \\
&\times P(\mathbf{x}_p | \mathbf{z}_{1:p}, E_p = 1, E_{p-1}).
\end{aligned}
\tag{35}
$$

where

$$
\begin{aligned}
P(\mathbf{x}_p | \mathbf{z}_{1:p}, E_p = 1, E_{p-1}) &= \frac{P(\mathbf{z}_p | E_p = 1, \mathbf{x}_p) P(\mathbf{x}_p | \mathbf{z}_{1:p-1}, E_p = 1, E_{p-1})}{P(\mathbf{z}_p | \mathbf{z}_{1:p-1}, E_p = 1, E_{p-1})} \\
&= \frac{l(\mathbf{z}_p | E_p = 1, \mathbf{x}_p) P(\mathbf{x}_p | \mathbf{z}_{1:p-1}, E_p = 1, E_{p-1})}{l(\mathbf{z}_p | \mathbf{z}_{1:p-1}, E_p = 1, E_{p-1})},
\end{aligned}
\tag{36}
$$

and

$$P(E_{p-1}|\mathbf{z}_{1:p}, E_p = 1) = \cfrac{\left[\begin{array}{c} P(\mathbf{z}_p|\mathbf{z}_{1:p-1}, E_p = 1, E_{p-1}) \times \\ P(E_p = 1|E_{p-1})P(E_{p-1}|\mathbf{z}_{1:p-1}) \end{array}\right]}{\sum\limits_{E_{p-1}\in\{0,1\}} \left[\begin{array}{c} P(\mathbf{z}_p|\mathbf{z}_{1:p-1}, E_p = 1, E_{p-1}) \times \\ P(E_p = 1|E_{p-1})P(E_{p-1}|\mathbf{z}_{1:p-1}) \end{array}\right]}$$
$$= \cfrac{\left[\begin{array}{c} l(\mathbf{z}_p|\mathbf{z}_{1:p-1}, E_p = 1, E_{p-1}) \times \\ P(E_p = 1|E_{p-1})P(E_{p-1}|\mathbf{z}_{1:p-1}) \end{array}\right]}{\sum\limits_{E_{p-1}\in\{0,1\}} \left[\begin{array}{c} l(\mathbf{z}_p|\mathbf{z}_{1:p-1}, E_p = 1, E_{p-1}) \times \\ P(E_p = 1|E_{p-1})P(E_{p-1}|\mathbf{z}_{1:p-1}) \end{array}\right]}. \tag{37}$$

The likelihood ratio in the formula is defined as:

$$l(\cdot) = \frac{P(\cdot)}{P(\mathbf{z}_p|E_p = 0)}. \tag{38}$$

From [37], We can obtain

$$l\left(z_p^{(n,k)}\middle|E_p = 1, \mathbf{x}_p\right) = I_0\left(\frac{z_p^{(n,k)}\left|I_p h^{(n,k)}(\mathbf{x}_p)\right|}{\sigma_v^2}\right)\exp\left(-\frac{\left|I_p h^{(n,k)}(\mathbf{x}_p)\right|^2}{2\sigma_v^2}\right), \tag{39}$$

where $I_0(\cdot)$ is the modified Bessel function with an order of zero [44]. The assumption is made that the noise in each bin is independent. Consequently, the complete likelihood function can be expressed as a product of all individual contributions from each bin.

$$l(\mathbf{z}_p|E_p = 1, \mathbf{x}_p) = \prod_{n=1}^{N}\prod_{k=1}^{\Psi\cdot M} l\left(z_p^{(n,k)}\middle|E_p = 1, \mathbf{x}_p\right). \tag{40}$$

The update step is shown in (36), and then the prediction step can be expressed as

$$P(\mathbf{x}_p|\mathbf{z}_{1:p-1}, E_p = 1, E_{p-1} = 1) = \int P(\mathbf{x}_p|\mathbf{x}_{p-1}, E_p = 1, E_{p-1} = 1)$$
$$\times P(\mathbf{x}_{p-1}|\mathbf{z}_{1:p-1}, E_{p-1} = 1)d\mathbf{x}_{p-1}, \tag{41}$$

and

$$P(\mathbf{x}_p|\mathbf{z}_{1:p-1}, E_p = 1, E_{p-1} = 0) = P(\mathbf{x}_p|E_p = 1, E_{p-1} = 0). \tag{42}$$

The probability of existence is

$$P(E_p = 1|\mathbf{z}_{1:p}) = \frac{P(\mathbf{z}_p|E_p = 1, \mathbf{z}_{1:p-1})P(E_p = 1|\mathbf{z}_{1:p-1})}{\sum\limits_{E_p\in\{0,1\}} P(\mathbf{z}_p|E_p, \mathbf{z}_{1:p-1})P(E_p|\mathbf{z}_{1:p-1})}$$
$$= \frac{l(\mathbf{z}_p|E_p = 1, \mathbf{z}_{1:p-1})P(E_p = 1|\mathbf{z}_{1:p-1})}{\sum\limits_{E_p\in\{0,1\}} l(\mathbf{z}_p|E_p, \mathbf{z}_{1:p-1})P(E_p|\mathbf{z}_{1:p-1})}$$
$$= \frac{l(\mathbf{z}_p|E_p = 1, \mathbf{z}_{1:p-1})P(E_p = 1|\mathbf{z}_{1:p-1})}{\left[\begin{array}{c} l(\mathbf{z}_p|E_p = 1, \mathbf{z}_{1:p-1})P(E_p = 1|\mathbf{z}_{1:p-1}) + \\ 1 - P(E_p = 1|\mathbf{z}_{1:p-1}) \end{array}\right]}. \tag{43}$$

where

$$P(E_p = 1 | \mathbf{z}_{1:p-1}) = P(E_p = 1 | E_{p-1} = 1) P(E_{p-1} = 1 | \mathbf{z}_{1:p-1}) \\ + P(E_p = 1 | E_{p-1} = 0) P(E_{p-1} = 0 | \mathbf{z}_{1:p-1}), \tag{44}$$

and

$$l(\mathbf{z}_p | E_p = 1, \mathbf{z}_{1:p-1}) = \sum_{E_{p-1} \in \{0,1\}} \left[ \begin{array}{c} l(\mathbf{z}_p | E_p = 1, E_{p-1}, \mathbf{z}_{1:p-1}) \\ \times P(E_{p-1} | E_p = 1, \mathbf{z}_{1:p-1}) \end{array} \right] \\ = \sum_{E_{p-1} \in \{0,1\}} \left[ \begin{array}{c} l(\mathbf{z}_p | E_p = 1, E_{p-1}, \mathbf{z}_{1:p-1}) \\ \times P(E_{p-1} | E_p = 1, \mathbf{z}_{1:p}) \end{array} \right]. \tag{45}$$

*3.4. Implementation of Improved Particle Filter*

Sequential importance sampling (SIS) can be used to approximate the $P(\mathbf{x_p} | \mathbf{z}_{1:p}, E_p = 1, E_{p-1})$ in (35), with an importance density function, $q(\mathbf{x}_{0:p} | \mathbf{z}_{1:p}, E_{0:p})$, which is the first-order Markov chain. Assume the important density can be decomposed as follows:

$$q(\mathbf{x}_{0:p} | \mathbf{z}_{1:p}, E_{0:p}) = q(\mathbf{x}_p | \mathbf{x}_{p-1}, \mathbf{z}_p, E_{0:p}) \times q(\mathbf{x}_{0:p-1} | \mathbf{z}_{1:p-1}, E_{0:p-1}). \tag{46}$$

The particle weights can be defined as

$$\omega_p = \frac{P(\mathbf{x}_{0:p} | \mathbf{z}_{1:p}, E_{0:p})}{q(\mathbf{x}_{0:p} | \mathbf{z}_{1:p}, E_{0:p})} \\ \propto \omega_{p-1} l(\mathbf{z}_p | \mathbf{x}_p, E_p) \frac{P(\mathbf{x}_p | \mathbf{x}_{p-1}, E_{p-1}, E_p)}{q(\mathbf{x}_p | \mathbf{x}_{p-1}, \mathbf{z}_p, E_{p-1}, E_p)}, \tag{47}$$

where $\propto$ denotes the proportionality.

If the importance density satisfies the following equations:

$$q(\mathbf{x}_p | \mathbf{x}_{p-1}, \mathbf{z}_p, E_{p-1}, E_p) = P(\mathbf{x}_p | \mathbf{x}_{p-1}, E_{p-1}, E_p), \tag{48}$$

the weights of particles can be expressed as

$$\omega_p^i \propto \omega_{p-1}^i l(\mathbf{z}_p | \mathbf{x}_p^i, E_p^i). \tag{49}$$

where *i* is the index of particle.

The normalized weights of particles are

$$\tilde{\omega}_p^i = \omega_p^i \left/ \sum_{i=1}^{K} \omega_p^i, \right. \tag{50}$$

*K* represents the number of particles.

Nevertheless, the SIS method frequently encounters the challenge of degeneracy. Degeneracy refers to the phenomenon in which only specific weights are consistently chosen after multiple iterations of SIS The resampling process can be employed to alleviate the degradation of the particles. There are several resampling methods, such as multinomial resampling, residual resampling, systematic resampling, etc. [45,46]. The systematic resampling is used in this paper and the pseudo-code is given in Algorithm 1.

The distribution $U(\cdot)$ in Algorithm 1 is uniform distribution and $\hat{\mathbf{x}}_p^i$ is the particles after resampling. $K_1$ and $K_2$ denote the quantities of particles prior to and subsequent to resampling, correspondingly. After resampling, the weight of all particles will be set to $1/K_2$.

---

**Algorithm 1** Systematic Resampling Pseudo-code.

$c_0 = \tilde{\omega}_p^0$
**for** $i = 1 : K_1 - 1$
   $c_i = c_{i-1} + \tilde{\omega}_p^i$
**end for**
$i = 0$
$u_0 \sim U\left(0, \frac{1}{K}\right)$
**for** $j = 0 : K_2 - 1$
   $u = u_0 + \frac{1}{K_2}$
   **while** $c_i < u$
      $i = i + 1$
   **end while**
   $\hat{\mathbf{x}}_p^i = \mathbf{x}_p^i$
   $\hat{\omega}_p^i = \frac{1}{K_2}$
**end for**

---

We can summarize the implementation steps of the proposed approach as follows:

1. Based on (48), generate $N_b$ newborn particles and $N_c$ continuing particles.

$$\begin{cases} \mathbf{x}_p^{(b)i} \sim P\left(\mathbf{x}_p \middle| \mathbf{x}_{p-1}, E_{p-1} = 1, E_p = 1\right) \\ \mathbf{x}_p^{(c)j} \sim P\left(\mathbf{x}_p \middle| E_{p-1} = 0, E_p = 1\right) \end{cases}. \tag{51}$$

where $i$ and $j$ represent the index of newborn and continuing particles, respectively. For the continuing particles, the target state-transition function can be obtained from (25). For the newborn particles, a uniform distribution can be used.

2. Calculate the weights of the particles and normalize them. For the newborn particles:

$$\begin{cases} \omega_p^{(b)i} = l\left(\mathbf{z}_p \middle| \mathbf{x}_p^{(b)i}, E_p^{(b)i} = 1\right) \\ \tilde{\omega}_p^{(b)i} = \omega_p^{(b)i} \middle/ \sum\limits_{i=1}^{N_b} \omega_p^{(b)i} \end{cases}. \tag{52}$$

For the continuing particles:

$$\begin{cases} \omega_p^{(c)i} = l\left(\mathbf{z}_p \middle| \mathbf{x}_p^{(c)i}, E_p^{(c)i} = 1\right) \\ \tilde{\omega}_p^{(c)i} = \omega_p^{(c)i} \middle/ \sum\limits_{i=1}^{N_c} \omega_p^{(c)i} \end{cases}. \tag{53}$$

3. Calculate $l\left(\mathbf{z}_p \middle| \mathbf{z}_{1:p-1}, E_p = 1, E_{p-1}\right)$.

$$\begin{aligned} &l\left(\mathbf{z}_p \middle| \mathbf{z}_{1:p-1}, E_p = 1, E_{p-1}\right) \\ &= \int l\left(\mathbf{z}_p \middle| \mathbf{x}_p, E_p = 1\right) \times P\left(\mathbf{x}_p \middle| \mathbf{z}_{1:p-1}, E_p = 1, E_{p-1}\right) d\mathbf{x}_p. \end{aligned} \tag{54}$$

We can use the Monte Carlo sampling method to calculate the integral, and the value is

$$l\left(\mathbf{z}_p \middle| \mathbf{z}_{1:p-1}, E_p = 1, E_{p-1}\right) = \begin{cases} \frac{1}{N_b} \sum\limits_{i=1}^{N_b} \omega_p^{(b)i}, E_{p-1} = 0 \\ \frac{1}{N_c} \sum\limits_{i=1}^{N_c} \omega_p^{(c)i}, E_{p-1} = 1 \end{cases}. \tag{55}$$

4. Calculate $P\big(E_{p-1}\big|\mathbf{z}_{1:p}, E_p = 1\big)$ in (37). For $E_{p-1} = 1$, define continuing mixing term.

$$
\begin{aligned}
\tilde{M}_c &= l\big(\mathbf{z}_p\big|\mathbf{z}_{1:p-1}, E_p = 1, E_{p-1} = 1\big)P\big(E_p = 1\big|E_{p-1} = 1\big)P\big(E_{p-1} = 1\big|\mathbf{z}_{1:p-1}\big) \\
&= \frac{1}{N_c}\sum_{i=1}^{N_c}\omega_p^{(c)i}\cdot(1 - P_d)P\big(E_{p-1} = 1\big|\mathbf{z}_{1:p-1}\big).
\end{aligned}
\tag{56}
$$

And for $E_{p-1} = 0$, the newborn mixing term is

$$
\begin{aligned}
\tilde{M}_b &= l\big(\mathbf{z}_p\big|\mathbf{z}_{1:p-1}, E_p = 0, E_{p-1} = 1\big)P\big(E_p = 1\big|E_{p-1} = 0\big)P\big(E_{p-1} = 0\big|\mathbf{z}_{1:p-1}\big) \\
&= \frac{1}{N_b}\sum_{i=1}^{N_b}\omega_p^{(b)i}\cdot P\big(E_{p-1} = 0\big|\mathbf{z}_{1:p-1}\big)P_b.
\end{aligned}
\tag{57}
$$

Then $P\big(E_{p-1}\big|\mathbf{z}_{1:p}, E_p = 1\big)$ can be calculated

$$
\begin{aligned}
M_c &= P\big(E_{p-1} = 1\big|\mathbf{z}_{1:p}, E_p = 1\big) \\
&= \frac{\tilde{M}_c}{\tilde{M}_c + \tilde{M}_b},
\end{aligned}
\tag{58}
$$

and

$$
\begin{aligned}
M_b &= P\big(E_{p-1} = 0\big|\mathbf{z}_{1:p}, E_p = 1\big) \\
&= \frac{\tilde{M}_b}{\tilde{M}_c + \tilde{M}_b}.
\end{aligned}
\tag{59}
$$

5. Calculate the probability of existence, $P\big(E_p = 1\big|\mathbf{z}_{1:p}\big)$ in (43).
   First, from (45), we can obtain

$$
l\big(\mathbf{z}_p\big|E_p = 1, \mathbf{z}_{1:p-1}\big) = \frac{M_c}{N_c}\sum_{i=1}^{N_c}\omega_p^{(c)i} + \frac{M_b}{N_b}\sum_{i=1}^{N_b}\omega_p^{(b)i}.
\tag{60}
$$

   Then, based on (43) and (44), the probability of existence can be calculated.

6. Calculate the posterior target state density from (35).

$$
\begin{cases}
\hat{\omega}_p^{(b)i} = M_b\tilde{\omega}_p^{(b)i} \\
\hat{\omega}_p^{(c)i} = M_c\tilde{\omega}_p^{(c)i}
\end{cases}.
\tag{61}
$$

   Combine the newborn particles set and the continuing particles set into a large set as follows:

$$
\Big\{\big(\mathbf{x}_p^{(t)i}, \hat{\omega}_p^{(t)i}\big)\big|i = 1, 2, \ldots, N_t; t = c, b\Big\}.
\tag{62}
$$

7. Resample $N_b + N_c$ particles down to the $N_c$ particles set, $\Big\{\big(\mathbf{x}_p^i, \frac{1}{N_c}\big)\big|i = 1, 2, \ldots, N_c\Big\}$.

8. First, Estimate the ambiguity number of the target.

$$
P\big(\eta_p\big|\mathbf{z}_{1:p}, E_p = 1\big) = \frac{\sum_{i=1}^{N_c}\Big[gt\big(\big(\tfrac{1}{2} + \eta_p\big)v_{\max}, v_p^i\big)\&gt\big(v_p^i, \big(-\tfrac{1}{2} + \eta_p\big)v_{\max}\big)\Big]}{N_c}
\tag{63}
$$

   where $gt(A, B)$ is the logical operation, when $A > B$ it equal to logic 1, in contrast, equal to logic 0. $v_p^i$ is the velocity of a particle with index $i$ and & is the logic AND operation.
   Then estimate the other motion state:

$$P\left(\mathbf{x}_p^i \middle| \mathbf{z}_{1:p}, E_p = 1\right) = \frac{\sum\limits_{i=1}^{N_c} \mathbf{x}_p^i \times \left[gt\left(\left(\frac{1}{2} + \eta_p\right)v_{\max}, v_p^i\right) \& gt\left(v_p^i, \left(-\frac{1}{2} + \eta_p\right)v_{\max}\right)\right]}{\sum\limits_{i=1}^{N_c} \left[gt\left(\left(\frac{1}{2} + \eta_p\right)v_{\max}, v_p^i\right) \& gt\left(v_p^i, \left(-\frac{1}{2} + \eta_p\right)v_{\max}\right)\right]} \tag{64}$$

## 4. Simulations and Results

### 4.1. Design of the Simulation System

In the simulation experiment, the echo signals are generated with the true parameters of a ubiquitous radar. The parameters of radar are shown in Table 1.

The echoes are produced by mixing the desired signal with zeros mean Gaussian white noise and can be expressed as

$$s_{echo}\left(\hat{t}, t_m\right) = s_1\left(\hat{t}, t_m\right) + w\left(\hat{t}, t_m\right), \tag{65}$$

where $w\left(\hat{t}, t_m\right)$ is the additive Gaussian white noise with zero mean and variance $\sigma_w^2$.

Then the SNR of the echoes is defined as

$$SNR_r = 10\lg\left(\frac{E}{\sigma_w^2}\right), \tag{66}$$

where $E$ is the power of the desired signal:

$$E = \int_{\hat{t}} s_1\left(\hat{t}, t_m\right)d\hat{t} \tag{67}$$

**Table 1.** Parameters of Radar.

| Parameter Name | Parameter Value |
| --- | --- |
| Pulse Repetition Frequency (PRF) | 5 kHz |
| Carrier Frequency | 1.36 GHz |
| Bandwidth | 4 MHz |
| Pulse Width | 2 us |
| Complex Sampling Frequency | 5 MHz |
| Integration Number | 2048 |
| Ambiguity Number | $\{-1, 0, 1\}$ |
| Range of Distance | $0 \sim 30$ km |
| Range of velocity | $-827.2 \sim 827.2$ m/s |

In the simulation, the target appears first in the sixth frame of ERD images and then disappears in the twenty-sixth frame. The motion state of the target is shown in Table 2. The initial radical velocity is $-300$ m/s, hence the corresponding ambiguity number is $-1$. As time progresses, the ambiguity number corresponding to the target velocity approaches 0.

**Table 2.** Motion State of Target.

| Parameter Name | Parameter Value |
| --- | --- |
| Initial Radical Distance | 15 km |
| Initial Radical Velocity | $-300$ m/s |
| Initial Radical Acceleration | $10$ m/s$^2$ |
| $q_1$ | 1 |
| $q_2$ | 5 |

Then the birth probability ($P_b$) and the death probability ($P_d$) are both set to 0.05. The prior density of birth particles follows the following uniform distributions: $r_0 \sim U(10,000 \text{ m}, 20,000 \text{ m})$, $v_0 \sim U(-500 \text{ m/s}, 500 \text{ m/s})$, $a_0 \sim U(-20 \text{ m/s}^2, 20 \text{ m/s}^2)$, $I_0 \sim U(30, 100)$. The numbers of particles are set to $N_b = 100,000$ and $N_c = 100,000$.

The Monte Carlo test times are set to 50 and the root mean square error (*RMSE*) is defined as

$$
\begin{cases}
RMSE(r_p) = \sqrt{\frac{1}{50} \sum\limits_{e=1}^{50} \left( \hat{r}_p^e - r_p \right)^2} \\
RMSE(v_p) = \sqrt{\frac{1}{50} \sum\limits_{e=1}^{50} \left( \hat{v}_p^e - v_p \right)^2} \quad , \\
RMSE(a_p) = \sqrt{\frac{1}{50} \sum\limits_{e=1}^{50} \left( \hat{a}_p^e - a_p \right)^2}
\end{cases}
\tag{68}
$$

where $\hat{r}_p^e$, $\hat{v}_p^e$ and $\hat{a}_p^e$ represent the estimation of the target's distance, velocity, and acceleration, respectively, at the frame $p$ in the $e$th Monte Carlo test. $r_p$, $v_p$, and $a_p$ denote the true distance, velocity, and acceleration, respectively.

To compare the performance of the LSM and PSM, we first calculate the weights based on the deviations of distance and velocity between the particles and real target, as Figure 5 shows. It indicates that given the same $SNR_r$, the weight of LSM is higher than that of PSM. Therefore, the integration efficiency of IPF is higher than traditional PF in the presence of DFM.

The ERD images with the various $SNR_r$ are shown in Figure 6. It is obvious that it is hard to detect the target using Constant False Alarm Rate (CFAR) technology.

To evaluate the effectiveness of the proposed method in this paper, we conducted four group experiments at different $SNR_r$ levels. The first experiment involved implementing the IPF on ERD images. In the second experiment, we utilized the traditional PF as proposed in [37], using the PSM as the spread function, on sub-RD images. The third experiment employed the method outlined in [37], again utilizing PSM as the spread function, on ERD images. Finally, the IPF was implemented on sub-RD images for the fourth experiment. These experimental methods are referred to as Method A, Method B, Method C, and Method D, respectively.

Figure 7 illustrates the estimation of the target existence probability for the three methods under various SNR. When the probability exceeds the detection threshold, the target can be considered alive at that moment. In this simulation, the value of the threshold is set to 0.7, which is denoted as a dotted line in Figure 7. It shows that due to the lack of consideration of VA, the energy of the target cannot be sufficiently accumulated, which results in the complete useless of Method B and Method D. Subsequently, a comparison between Method A and Method C reveals that the IPF exhibits superior performance in the presence of DFM within ERD imagery when contrasted with the conventional PF. Although the performance of traditional PF can reach the IPF under relatively high $SNR_r$. LSM can achieve far better performance than PSM under low $SNR_r$.

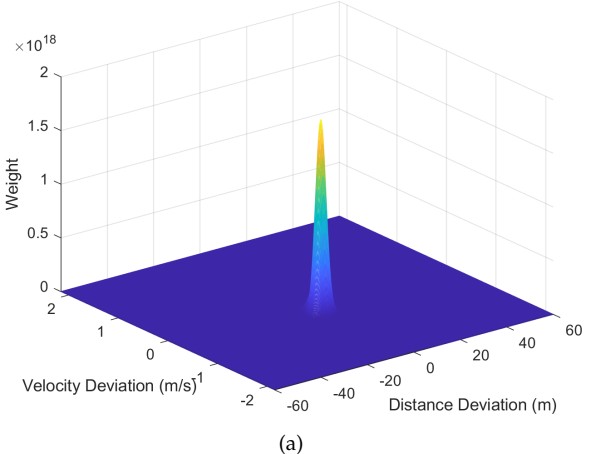

(a)

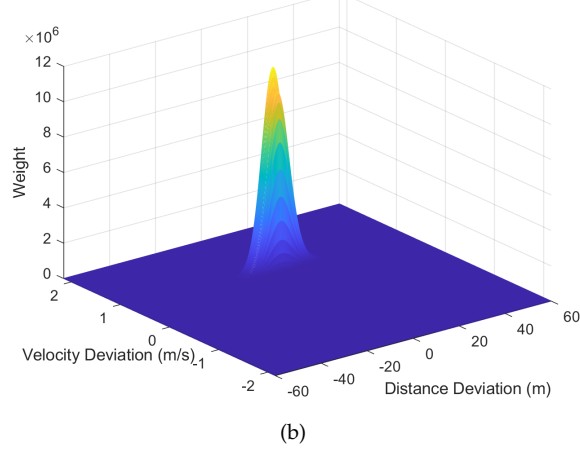

(b)

**Figure 5.** *Cont.*

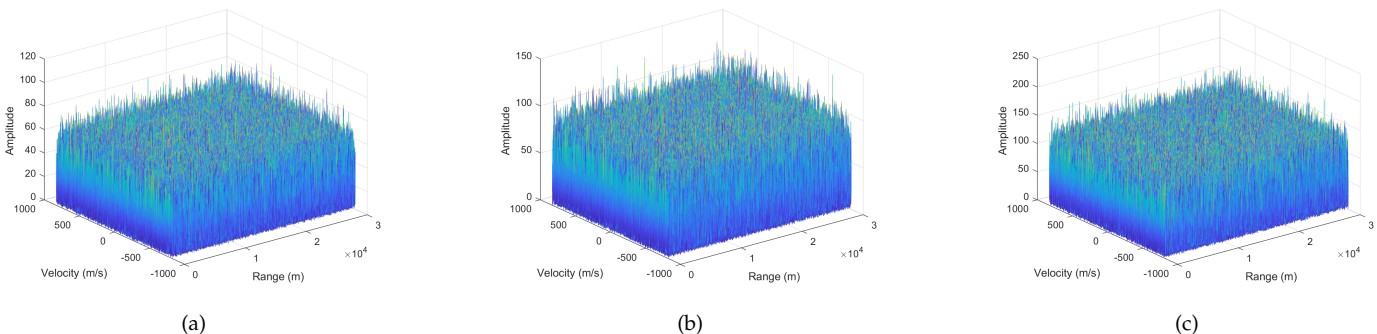

**Figure 5.** Relationship Between Weights and the Deviations. (**a**) LSM with $SNR_r$ −15 db. (**b**) PSM with $SNR_r$ −15 db. (**c**) LSM with $SNR_r$ −18 db. (**d**) PSM with $SNR_r$ −18 db. (**e**) LSM with $SNR_r$ −21 db. (**f**) PSM with $SNR_r$ −21 db.

**Figure 6.** (**a**) ERD image with $SNR_r$ −15 db. (**b**) ERD image with $SNR_r$ −18 db. (**c**) ERD image with $SNR_r$ −21 db.

RMSE reflects the deviation between the estimated data value and the true value. The RMSEs of Method A and Method C with different $SNR_r$ are shown in Figure 8. With the decline of $SNR_r$, the tracking accuracy of the two methods degrades. Meanwhile, Method A shows a better performance than Method C, especially under low $SNR_r$. It indicates that the

distance and velocity of the target can be estimated more accurately by Method A. The RMSE of the two methods increases as the $SNR_r$ decreases.

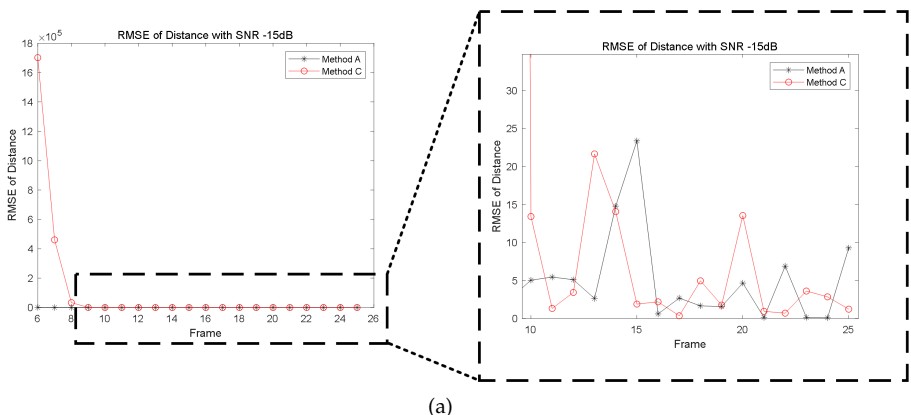

**Figure 7.** Estimation of the Target Existence Probability. (**a**) With $SNR_r$ −15 db. (**b**) With $SNR_r$ −18 db. (**c**) With $SNR_r$ −21 db. (**d**) With $SNR_r$ −24 db.

**Figure 8.** *Cont.*

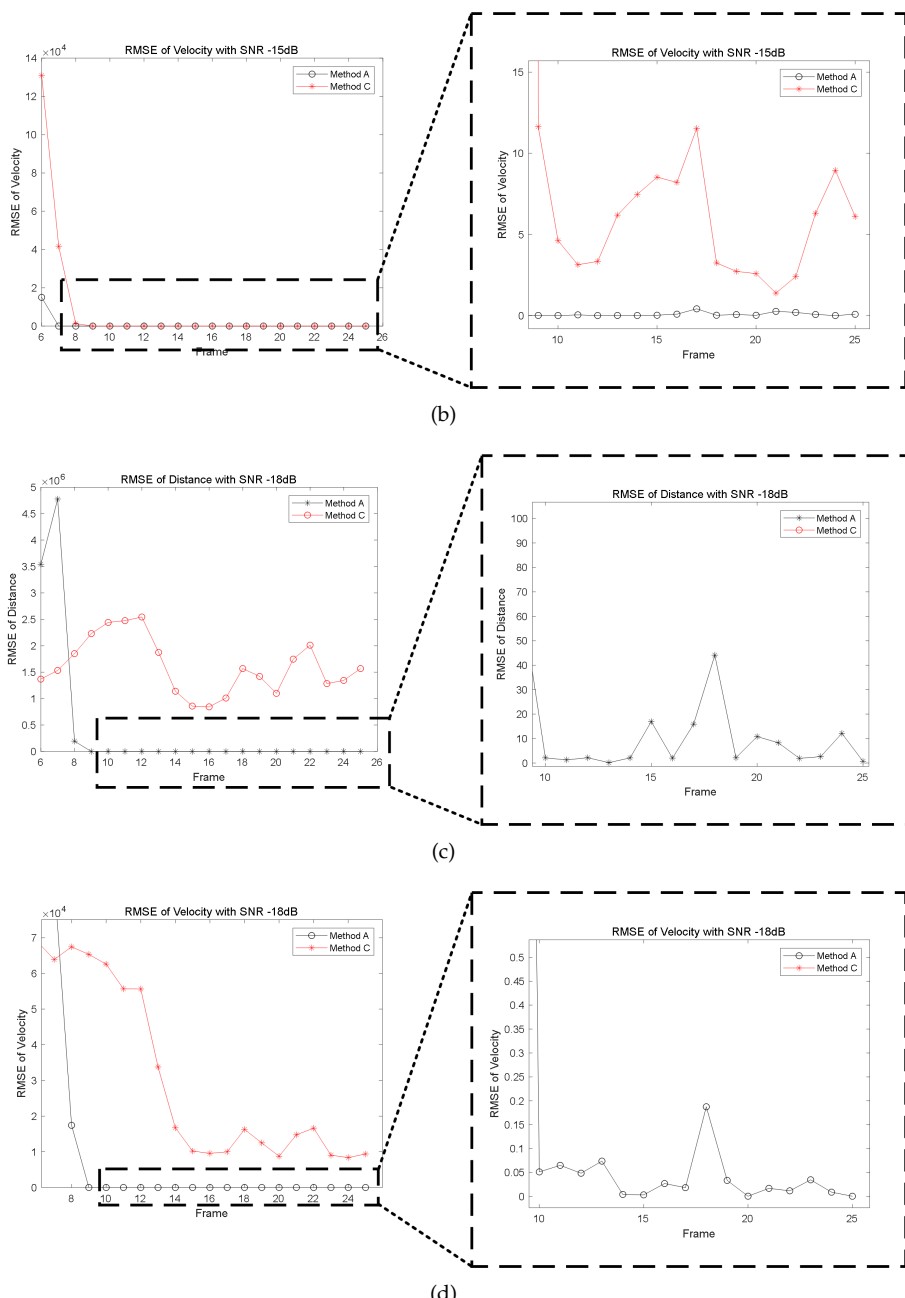

**Figure 8.** The RMSE of: (**a**) Distance with $SNR_r$ −15 db. (**b**) Velocity with $SNR_r$ −15 db. (**c**) Distance with $SNR_r$ −18 db. (**d**) Velocity with $SNR_r$ −18 db.

### 4.2. Integration Efficiency Analysis

The proposed KT-MFP-IPF-TBD algorithm in this paper is a hybrid integration algorithm, which can achieve integrated detection and tracking. However, the target integration efficiency remains an important reference for evaluating its algorithmic performance. The detection capability of an algorithm is closely related to its integration efficiency, hence detection capability can be used to evaluate integration efficiency. This paper compared the detection capability of three methods: A hybrid integration method proposed in [16], which is consisted of moving target detection and generalized radon transform (MTD-GRT); A coherent integration method proposed in [8], which is consisted of second-order Keystone transform, Fractional Fourier Transform and radon Fourier transform (SoKT-FrFT-RFT); KT-MFP-IPF-TBD method proposed in this paper. For the methods proposed in papers [8,16], a target detector is also required to detect the targets. In this paper, the Cell Average Constant

False Alarm Rate (CA-CFAR) is adopted as the detector [47]. The number of target's echoes is 20,480, the three methods perform detection based on these echoes. To prevent RM and DFM, the coherent integration number of MTD in the MTD-GRT method is set to 256, therefore the GRT is used to perform incoherent integration among 80 sub-RD images. In the SoKT-FrFT-RFT method, all of the 20,480 echoes are used in coherent integration. In the KT-MFP-IPF-TBD method, the coherent integration number is set to 2048, therefore the IPF is used to perform incoherent integration among 10 ERD images. The False alarm probability of CA-CFAR is set to $10^{-7}$. Performing 1000 times Monte Carlo tests, The experiment results are shown in Figure 9. The computational complexity of the three methods is shown in Table 3, where flops are used to represent complexity.

From the experiment result, it can be seen that KT-MFP-IPF-TBD, as a hybrid accumulation approach, has a detection performance very close to the coherent accumulation approach SoKT-FrFT-RFT. Specifically, for the needed 80% detection probability, the needed SNR of the SoKT-FrFT-RFT method is −22.5 dB and that of KT-MFP-IPF-TBD is –21.5 dB, i.e., the proposed method only suffer from 1 dB SNR loss. However, the computational complexity of KT-MFP-IPF-TBD is two orders of magnitude lower than that of SoKT-FrFT-RFT. For the needed 80% detection probability, the needed SNR of the MTD-GRT method is −17.5 dB. Hence, as with MTD-GRT, which is also a hybrid integration method, its detection performance is significantly lower than that of KT-MFP-IPF-TBD and has a 4 dB higher SNR loss compared to method KT-MFP-IPF-TBD.

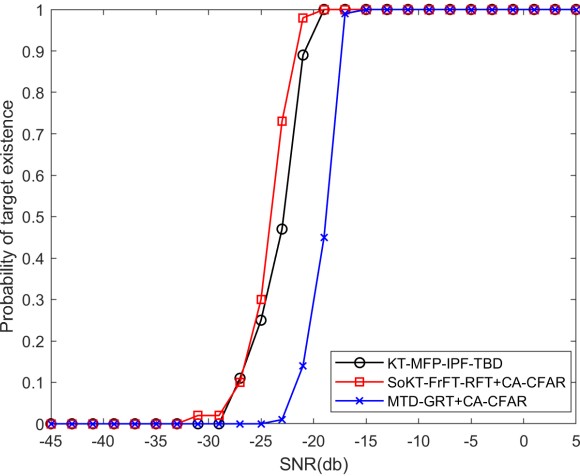

**Figure 9.** Detection capability of the three methods.

**Table 3.** Calculation complexity of the three methods.

| Method Name | Complexity (Flops) |
| --- | --- |
| KT-FrFT-RFT+CA-CFAR | $2.4785 \times 10^{12}$ |
| MTD-GRT+CA-CFAR | $1.3571 \times 10^{9}$ |
| KT-MFP-IPF-TBD | $2.0547 \times 10^{10}$ |

### 4.3. Ubiquitous Radar Actual Data Validation

In this paper, the algorithms were validated using holographic radar echo data from high-speed and highly maneuverable aircraft targets. The aircraft hovers in the air with a maximum speed exceeding 275 m/s, resulting in VA and ambiguity number values of −1, 0, and 1. The ERD plane is composed of three sub-RD and 200 frames of turning data are selected for experimentation. The ERD plane under two different motion states is shown in Figure 10. It can be observed that when the acceleration is low, the target energy is more concentrated and the SNR is higher; when the acceleration is high, the target energy is scattered and the SNR is lower. The estimation results of the target state are shown in

Figure 11, it can be seen that the proposed method in this paper is effective in detecting and tracking high-speed and highly maneuverable targets.

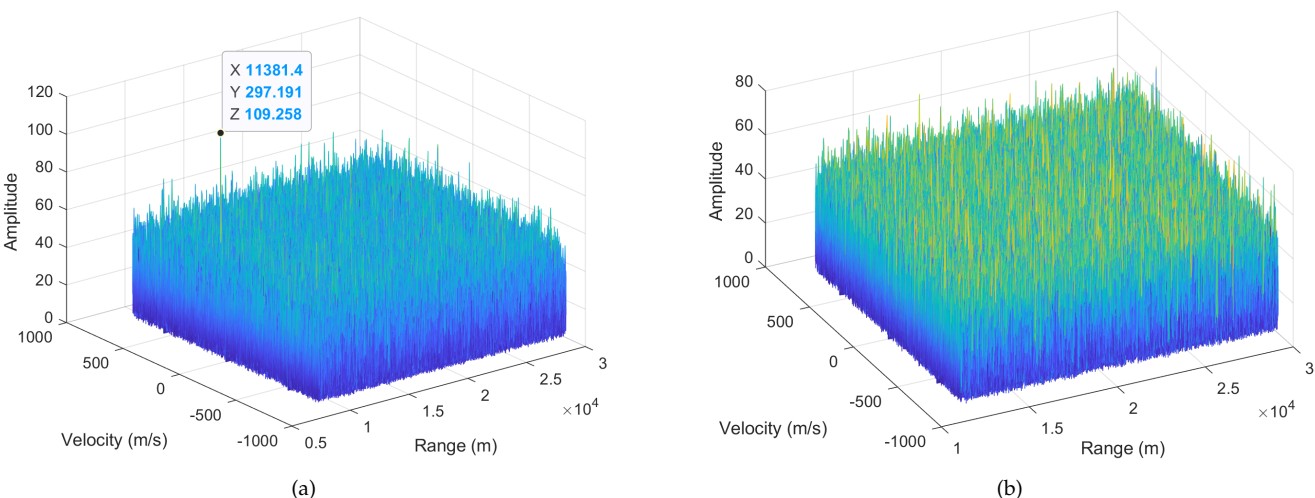

**Figure 10.** ERD images of aircraft. (**a**) With lower acceleration. (**b**) With higher acceleration.

**Figure 11.** Estimation of aircraft. (**a**) Probability of target existence. (**b**) Radial range. (**c**) Radial velocity. (**d**) Radial acceleration.

## 5. Conclusions

Due to the low transmission gain of ubiquitous radar, the LTCA is a key technique for ubiquitous radar signal processing. However, both RM and DFM can deteriorate the actual performance of this technique. To compensate for RM and DFM, this paper proposes a KT-MFP-IPF-TBD method compatible with ubiquitous radar. First, we apply the KT to mitigate the effects of first-order RM. Subsequently, we employ the MFP to resolve the VA. This sequence of steps yields an ERD image that encompasses a broader range of Doppler frequencies. The ubiquitous radar system, known for its comprehensive coverage in both space and time domains, enables the utilization of recursive TBD algorithms for improved performance. To this end, we propose an IPF incorporating an LSM for the purpose of target tracking and detection.

In our simulation, with a total number of 200,000 particles, the final results showed that this method can detect and track targets reliably and presents a better performance than the traditional PF. Compared to the SoKT-FrFT-RFT method, the proposed approach in this paper can reduce computational complexity by two orders of magnitude with only a 1 dB loss in signal-to-noise ratio. Moreover, compared with other hybrid integration methods, the detection performance of KT-MFP-IPF-TBD is significantly higher resulting in a 4dB greater SNR improvement. Finally, we validated the proposed method by utilizing empirical data from high-speed and high-maneuver targets. The results demonstrate that the algorithm put forth in this study yields excellent efficacy.

**Author Contributions:** Conceptualization, X.P.; methodology, X.P.; software, X.P.; validation, X.P., W.W. and Y.Z.; formal analysis, X.P. and Q.S.; investigation, X.P.; resources, X.P. and Y.Z.; data curation, X.P. and Y.Z.; writing—original draft preparation, X.P.; writing—review and editing, X.P., W.W., Q.S. and Y.Z.; visualization, X.P. and Y.Z.; supervision, X.P. and Y.Z.; project administration, X.P., W.W., Q.S. and Y.Z.; funding acquisition, Y.Z. All authors have read and agreed to the published version of the manuscript.

**Funding:** This work was supported by the Shenzhen Fundamental Research Program (No.JCYJ20180 307151430655).

**Conflicts of Interest:** The authors declare no conflict of interest.

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
