# Peer review of "An Hybrid Integration Method-Based Track-before-Detect for High-Speed and High-Maneuvering Targets in Ubiquitous Radar"

_remotesensing, doi:10.3390/rs15143507_

Round 1
Reviewer 1 Report
Please see the attatched file.

Minor editing of English language required. Please see the first comment.
Reviewer 2 Report
This paper presents a hybrid integration method combining incoherent and coherent integration. The coherent integration method of (Keystone-transform and matched filtering processing) KT-MFP method is used to correct (range migration) RM and generate an extended ( range- Doppler) RD map. Then, the IPF-TBD method is utilized to compensate for DFM and carry out incoherent integration on the extended RD map. Simulation results verified the effectiveness of the proposed algorithm.
In all, the manuscript is well-organized and easy to understand. The study of the hybrid integration method combining incoherent and coherent for ubiquitous radar is of practical importance for promoting the engineering applications of integration methods. However, the manuscript contains some errors in detail, especially in some equations and simulation figures. The following issues should be more thoroughly explained before it can be accepted for publication.
--In the introduction section, the authors state that “This feature determines that batch algorithms can not be adapted to a maneuvering target.” To my knowledge, some recent research works have been published on batch algorithms to solve target maneuvering and nonlinear transformation problems, e.g.
[r1] Yi et al, Multi-Frame Track-Before-Detect Algorithm for Maneuvering Target Tracking, IEEE TVT, 2020.
[r2] Wang et al, Pseudo-Spectrum Based Track-Before-Detect for Weak Maneuvering Targets in Range-Doppler Plane, IEEE TVT, 2021.
[r3] Li et al, Adaptive Multiframe Detection Algorithm With Range-Doppler-Azimuth Measurements, IEEE TGRS, 2022.
Thus, the literature review can be improved by including some recent works.
-- I’m a bit concerned about the correctness of the eq.(29) and eq.(32). Specifically, the probability transition matrix eq.(29) may be incorrect, as typically the sum of each row should equal one. Furthermore, how does the spread function eq.(32) compensate DFM, and why add this phase term? Please the authors to clarify and review.
--In Section 3.3, it would be good if the authors can further provide the derivation of eq.(39), which can help reviewers easily follow.
-- I'm a bit concerned about the presentation of the notation. I found some symbols are over-complicated. For example, in eq. (24), the state vector is denoted by x_p. Later, the authors further defined x_p to indicate logical operation. Therefore, please the authors to check and modify.
--The caption of figure.6(a) and figure.6(b) are the same.
--Finally, the proposed method should be compared with existing approaches of hybrid integration in the simulation section.
Needs improvement
Reviewer 3 Report
The topic is actual, and the introduction is interesting for a reader. However, the simulation conditions are not well discussed. The proposed approach was illustrated only on some specific simulations, which is not enough to draw a complete and accurate conclusion about the proposed approach. Also, the experiments are always carried out for only one target, and therefore the conclusion should discuss one „target“ and not „targets“. Are your algorithms prepared for multitarget situations?
For specific remarks, see the attachment.

Two mistakes even in the title! (A hybrid... based on...) You can use freely available software (e.g. Grammarly) for correcting your text. Many mistakes like "complexity" instead of complex, "numbers is" instead of are, "is closed to" instead of close, "a initial" instead of an, "target existence are separated" instead of is... The use of articles (a/an, the) also needs to be improved.
Round 2
Reviewer 2 Report
The authors have revised the manuscript well. The paper can be accepted now.
The authors can go over the paper to check the English languages to avoid grammar issues.